# Naphthalene Phthalimide Derivatives as Model Compounds for Electrochromic Materials

**DOI:** 10.3390/molecules28041740

**Published:** 2023-02-11

**Authors:** Magdalena Zawadzka, Paweł Nitschke, Marta Musioł, Mariola Siwy, Sandra Pluczyk-Małek, Damian Honisz, Mieczysław Łapkowski

**Affiliations:** 1Faculty of Chemistry, Silesian University of Technology, Strzody 9, 44-100 Gliwice, Poland; 2Centre for Organic and Nanohybrid Electronics, Silesian University of Technology, Konarskiego 22B, 44-100 Gliwice, Poland; 3Centre of Polymer and Carbon Materials, Polish Academy of Sciences, M. Curie-Skłodowska 34, 41-819 Zabrze, Poland

**Keywords:** electrochemistry, spectroelectrochemistry, electrochromism, phthalimides

## Abstract

Electrochromism of organic compounds is a well-known phenomenon; however, nowadays, most research is focused on anodic coloring materials. Development of efficient, cathodic electrochromic materials is challenging due to the worse stability of electron accepting materials compared with electron donating ones. Nevertheless, designing stable cathodic coloring organic materials is highly desired—among other reasons—to increase the coloration performance. Hence, four phthalimide derivatives named 1,5-PhDI, 1,4-PhDI, 2,6-PhDI and 3,3′-PhDI were synthesized and analyzed in depth. In all cases, two imide groups were connected via naphthalene (1,5-PhDI, 1,4-PhDI, 2,6-PhDI) or 3,3′-dimethylnaphtidin (3,3′-PhDI) bridge. To observe the effect of chemical structure on physicochemical properties, various positions of imide bond were considered, namely, 1,5- 1,4- and 2,6-. Additionally, a compound with the pyromellitic diimide unit capped with two 1-naphtalene substituents was obtained. All compounds were studied in terms of their thermal behavior, using differential calorimetry (DSC) and thermogravimetric analysis (TGA). Moreover, electrochemical (CV, DPV) and spectroelectrochemical (UV–Vis and EPR) analyses were performed to evaluate the obtained materials in terms of their application as cathodic electrochromic materials. All obtained materials undergo reversible electrochemical reduction which leads to changes in their optical properties. In the case of imide derivatives, absorption bands related to both reduced and neutral forms are located in the UV region. However, importantly, the introduction of the 3,3′-dimethylnaphtidine bridge leads to a noticeable bathochromic shift of the reduced form absorption band of 3,3′-PhDI. This indicates that optimization of the phthalimide structure allows us to obtain stable, cathodic electrochromic materials.

## 1. Introduction

Electrochromic materials are able to change their color hue upon applied potential, due to redox processes [1]. Such features might be applied in various devices, such as smart windows, auto-dimming car mirrors, adaptive camouflage, wearable displays, data storage, energy storage or e-paper [2,3,4,5,6,7,8]. Among various classes of organic materials investigated for this effect, such as metal complexes [9] or coordination polymers [10,11], conjugated polymers and small molecules seem to be particularly interesting [6,12]. They exhibit high coloration efficiency, vivid colors at low potential range and durable and fast switching [13]. Moreover, color tunability of organic molecules, through structural modification, might be easily achieved, which is another important feature of this class of compounds. Depending on the redox process, as a result of which color hue appears, electrochromic materials may be divided into anodically colored (reversible oxidation) or cathodically colored (reversible reduction). Most research is focused on the development of novel anodically colored materials, consisting of electron-donating groups such as triarylamines, thiophene or carbazole [14,15,16]. Materials which have an electrochromic effect, resulting from reversible reduction, draw much less interest, on account of the charge trapping effect of H_2_O/O_2_—complex [17]. Development of efficient, cathodically colorable materials would, however, be highly desired to increase the coloration performance through reduction of energy consumed by cathodes, when utilized together with anodically coloring material for device fabrication.

The most promising materials for cathodic coloring are arylene diimides (DI), which are n-type semiconductors. The number of reports regarding electrochromic diimides is, however, scarce [18,19,20,21,22] due to their limited solubility required for thin film formation on the transparent electrode. This could be overcome by various strategies, such as electropolymerization which leads to deposition of the active film directly on the working electrode surface [19,20]. Another approach relies on in situ growth of a metal–organic framework on the substrate [21,22]. Finally, a soluble poly (amylic acid) might be deposited on the electrode and subsequently imidized under thermal treatment [23]. Most of the interest is focused on novel naphthalene (NDI) and perylene diimides (PDI) due to their superior photophysical properties [24]. Pyromellitic diimides (PMDI), as well as phthalimides (PhDI), have been almost completely omitted as a building block when developing new electrochromic compounds. Such units might provide a better solubility compared to NDI or PDI. There has already been a report presented which describes the use of PMDI-based salts for construction of a dual-mode photochromic and intrinsically electrochromic device [25].

In this work, we focus on phthalimides, being model compounds for polymers with PMDI building blocks. According to our best knowledge, when it comes to electrochromic cathodic materials, such units have virtually not been investigated. Hence, we decided to study, in depth, the impact of chemical structure of some model PhDI and PMDI derivatives on their properties, especially on their electrochemical reduction and the products of this process. For these purposes, PhDI units were coupled with naphthalene moieties, due to their advantageous properties [24]. To observe the effect of chemical structure on physicochemical properties, various positions of imide bond were considered, namely 1,5- 1,4- and 2,6-. Moreover, a compound with the pyromellitic diimide unit was obtained, capped with two 1-naphtalene substituents. Finally, a compound with 3,3′-dimethylnaphtidine was obtained, due to the promising properties of polyimide consisting of such a building unit, as reported in [23]. All of the compounds were studied in terms of their thermal behavior, using differential calorimetry (DSC) and thermogravimetric analysis (TGA). Absorption measurements (UV–Vis) of their solutions, as well as electrochemical and spectroelectrochemical (UV–Vis and EPR) analyses, were performed to evaluate the obtained materials in terms of their application as cathodic electrochromic materials.

## 2. Results

All of the studied imides were obtained by solution condensation of various anhydrides and amines in dimethylacetamide (DMA). The chemical structures, together with detailed synthetic protocols, as well as ^1^H-NMR and FTIR characterization data, are given in the experimental section, while the registered spectra are presented in Appendix A (see Appendix A for ^1^H-NMR and FTIR spectra, respectively).

### 2.1. Structural and Solubility Studies

The imides were obtained in good yields, and of high purity (good agreement between calculated and found elemental composition). In any of the registered NMR spectra neither an amine nor a carboxylic proton could be found. This supports reaction occurrence and excludes the formation of amylic acids. Proton spectra of 3,3′-PhDI and 1-PMDI are well resolved and the number of signals together with their integrals are in a very good agreement with the number of aromatic protons. The spectra of 1,5-PhDI, 1,4-PhDI and 2,6-PhDI are worse resolved, probably due to their worse solubility; however, the number of signal groups and their integrals also supports the expected chemical structure. The FTIR spectra of all compounds revealed characteristic absorption bands assigned to either asymmetric or symmetric C=O stretching in ranges of 1781–1776 cm^−1^ or 1727–1716 cm^−1^, respectively. Solubility of obtained imides was investigated in several solvents (Table 1).

All of the synthesized compounds were very well soluble in DMF and NMP. In DMSO, imides based on naphthalene core (1,5-PhDI; 1,4-PhDI and 2,6-PhDI) are required to be heated to dissolve, contrary to compounds synthesized from naphthalene monoamine (1-PMDI) or methyl-substituted diaminonaphtidine (3,3′-PhDI). Moreover, the presence of methyl substituents in 3,3′-PhDI also provides a solubility in much less polar THF.

### 2.2. Thermal Properties

The glass transition temperatures (*T_g_*) were designated using differential scanning calorimetry (DSC—Appendix A) and thermogravimetric analyses (TGA—Appendix A), respectively. Almost all (except 2,6-PhDI) DSC curves registered during the first heating stage did not reveal endotherms related to melting, suggesting an amorphous character of the samples. Simultaneously a characteristic inflection on the curves was observed, which was ascribed to the glass transition. The temperatures of *T_g_*, taken from the midpoint of the inflection, are gathered in Table 2.

Imide based on 2,6-diaminonaphtalene (2,6-PhDI) during the first heating stage showed a double endotherm at 451 °C and 484 °C connected with melting, after which decomposition of material took place (Appendix A). The glass transition temperature was registered during the second heating stage for a sample annealed to the temperature of the first endotherm and quenched. No mass loss was registered.

All of the studied imides showed high glass transition temperatures. The highest *T_g_* was observed for 1-PMDI, consisting of two naphthalene moieties (Table 2). This might result from the most rigid structure of this imide. Slightly lower *T_g_* was registered for compounds based on 1,5-naphtalene (1,5-PhDI) and 3,3′-dimanonaphtidine (3,3′-PhDI). Interestingly, a change of imide substituent’s position from 1,5- to 1,4- or 2,6- resulted in the lowering of the glass transition temperature, which could be a result of coplanarity deterioration along two aromatic moieties, weakening the π-π stacking.

### 2.3. Optical Properties

The absorption of synthesized imides was recorded for 10^−4^ M solutions using DMF as a solvent. Registered spectra (Figure 1) showed a rather narrow absorption range in near ultraviolet, which is characteristic for core-unsubstituted naphthalene diimides [26,27].

Electronic spectra of all compounds revealed a single absorption band, connected with electron transitions between π→π * levels. The position of this band was identical (291 nm) for solutions of imides based on 1,5-naphtalene (1,5-PhDI) and 1,4-naphtalene (1,4-PhDI). A change of imide bonds positions to 2,6- (2,6-PhDI) resulted in a slight bathochromic shift of the lowest energy absorption band to 292 nm. Comparing these spectra with spectra of core substituted phthalimides, it can be noticed that N-substituted ones exhibit absorption at lower wavelengths in the UV region [28,29]. This can be advantageous due to the fact that it can result in electrochromic materials with colorless and colored states.

The spectra of imide 1-PMDI was characterized by a slightly pronounced vibronic structure. The energy peak corresponding to 0–1 transition, localized at 285 nm, was of the highest energy and was accompanied by peak of lower energy 0–0 (294 nm) and of higher energy (0–2), visible as a deflection at 273.0 nm [30,31,32]. The π→π * absorption band of 3,3′-PhDI was localized at the highest wavelengths (297 nm), suggesting enhanced *π*-conjugation area. Even though this could be attributed to the presence of additional naphthalene moiety, it is known that due to the twist of two aromatic units, a deterioration of co-planarity might take place, reducing the effective *π*-conjugation area [33]. Thus, the observed bathochromic shift is probably a result of substitution with methyl groups of a slight electron-donating character. Moreover, an additional well-pronounced deflection is visible on the absorption edge at 324 nm, which could be attributed to the charge transfer band between acceptor naphthalene diimide and capping phenyl groups [34].

### 2.4. Electrochemical Behavior

All investigated imides underwent reversible or quasi-reversible electrochemical reduction. Reduction of 1-PMDI proceed as a two-step process, whereas the rest of the analyzed imides showed a one-step electrochemical reduction (Figure 2 and Appendix A, Table 3). The CV voltammogram of 1-PMDI (Figure 2a) corresponds to a known mechanism of arylene diimides reduction [35,36,37,38]. In the first stage, a radical anion is generated; in the second stage, dianion. The minimum of cathode peak associated with the first stage of 1-PMDI reduction was observed at −1.21 V, and the potential of the corresponding anode peak was −1.10 V. In turn, for the second step, the potential of the minimum of the cathode peak was registered at −1.86 V and the corresponding anode peak was observed at −1.75 V. The remaining compounds exhibited one redox couple in a similar manner, such as in the case of compounds with only one imide group, indicating the fact that the reduction of both phthalimide units occurs concomitantly at the same potential; hence, there is no interaction between them.

Among all investigated compounds, 1-PMDI underwent reduction at the highest value of potential, which means that it is easiest to reduce. This indicates the fact that pyromellitic diimide exhibits stronger electron accepting properties compared with the phthalimide unit. Considering the reduction potentials of the investigated imides, 3,3′-PhDI is the easiest to reduce. Potential for the reduction of investigated compounds decreases (is shifted to more negative potentials) in order: 1-PMDI (−1.22 V) > 3,3′-PhDI (−1.77 V) > 1,4-PhDI (−1.90 V) > 2,6-PhDI (−1.93 V) > 1,5-PhDI (−1.96 V). Hence, the introduction of the 3,3′-dimethylnaphtidine unit between the imide group leads to the noticeable increase in the reduction potential compared with naphthalene unit, confirming the promising properties of polyimide consisting of such a building unit.

### 2.5. EPR Spectroelectrochemistry

Electrochemical reduction of all investigated derivatives leads to radical anion generation, which was confirmed by EPR spectroelectrochemical measurement. In the case of radical anion of 1-PMDI, which is the product of the first step of the 1-PMDI reduction, the EPR spectrum consisted of eleven spectral lines (Figure 3a). Such a hyperfine structure can be fitted assuming isotropic hyperfine interactions of the unpaired electron with the nuclei of two nitrogen and two hydrogen atoms of pyromellitic diimide, indicating the lack of interaction of the radical with N-substituents. The lowering of working electrode potential to the value corresponding to the second reduction step 1-PMDI led to the decrease in the intensity of the EPR signal. This is associated with the formation of spinless dianions. Registered EPR spectra of radical anions of all analyzed phthalimides are similar, showing hyperfine structure (Figure 3b and Appendix A) which can be simulated assuming isotropic hyperfine interactions of the unpaired electron with the nuclei of one nitrogen and four hydrogen atoms of phthalimide units (Table 4). Radicals are characterized by the same g-factor, which additionally confirms that in the case of all studied phthalimides, reduction takes place on the same structural unit. EPR measurements clearly show that independently from the bridge between imide groups, the obtained radical anions localize only on phthalimide units. The simulated spectrum of 3,3′-PhDI radical anion is characterized by broader linewidth due to the fact that it is less resolved compared with the rest of phthalimides. This can be associated with the different rate of relaxation of this radical compared with the rest of the radicals.

### 2.6. UV–Vis Spectroelectrochemistry

To evaluate the potential electrochromic properties of investigated compounds, a UV–Vis spectroelectrochemical experiment was applied (Figure 4a,b). The neutral form of each of the tested compounds in DMF-based electrolyte was characterized by the presence of a single absorption band (see Section 2.3 Optical properties). The experiments were repeated using DCM-based electrolyte as well (as DMF substitute) and the UV–Vis spectra were recorded again. The negligible impact of the solvent was observed; absorption maxima of individual bands were observed at similar wavelengths as for DMF. For 1-PMDI, the wavelength of absorption maximum was observed at 288 nm, for 1,5-PhDI, at 291 nm, for 2,6-PhDI, at 292 nm and for 1,4-PhDI, at 289 nm. Compounds were further electrochemically reduced using a potential range from 0 to −2 V; during the measurement, the potential of the working electrode was gradually lowered. In the case of 1-PMDI, UV–Vis spectroelectrochemical results showed two-stage character of the pyromellitic diimide derivative reduction (Figure 4a) The first step led to a rise of low intensity band with the maximum at 717 nm. The gradual increase of this peak was observed in the potential range from −0.9 to −1.2 V and is associated with the formation of radical anions. Further decreasing of the working electrode potential led to bleaching of this band and the appearance of an absorption band of higher intensity with a maximum at 552 nm, which is a result of the dianions generation. In the literature, there have been already reported results associated with UV–Vis spectroelectrochemical analysis of the PMDI derivative with triphenylamine units as N-substituents [35]; however, in the case of 1-PMDI, the electrochromic effect is more evident. What is more, in the case of the previously reported PMDI derivative, its reduction proceeds as a one-step process due to the fact that the surplus electron density in the formed radical anion could not be efficiently delocalized [35]. Evidently, the change of N-substituents has led to better distribution of electron density which results in two-step reduction of 1-PMDI. This provides the multi-electrochromic material, as absorption bands associated with radical anion and dianion are located at different wavelengths.

For all investigated phthalimide derivatives, at a potential of about −1.6 V, a decrease in the intensity of the band assigned to the neutral form was noticed. Simultaneously, the appearance of a new band was observed, the intensity of which was increasing, while the potential value was decreasing. The maximum of absorption band related to the reduced form was registered at 340 nm for 1,4-PhDI (Figure 4b), 337 nm for 2,6-PhDI, 336 nm in the case of 1,5-PhD and at 340 nm for 3,3-PhDI (Appendix A and Table 5). The presence of this band can be attributed to the formation of the radical anions of the compounds.

### 2.7. Theoretical Calculation

To deepen the analysis and confirm our findings, the DFT and TDDFT calculations were conducted. The theoretical HOMO and LUMO energies are given in Table 6. The values of LUMO levels are different from the values of EA calculated based on CV (Table 3) measurements. It is worth to note that, although DFT calculations are useful to model orbital energies of organic electroactive materials, differences between theory and experiment exist often. HOMO energies obtained from theoretical calculations are rather considered consistent with the experiment and can be used to predict oxidation potentials. On the other hand, the calculation of LUMO energies very often gives values that are less negative compared with what is experimentally determined [39,40], such as in our case. The reason for such discrepancy is the lack of an electron in this orbital [39]. Nevertheless, in the case of theoretical calculation, as well as experimental results, the 1-PMDI molecule is characterized by the lowest value of the LUMO and EA energy, which shows again that pyromellitic diimide exhibits stronger electron accepting properties compared with the phthalimide unit and it is a promising building block for cathodically colored stable electrochromic materials. The LUMO energy values of the rest of the investigated compounds are similar (around ca. −2.50 eV), showing that this value is little dependent on the bridge between imide groups. This is in agreement with the electrochemical results, which show that the bridge between phthalimide units has little impact on reduction potential. This is directly connected with the fact that for all investigated molecules, HOMO and LUMO orbitals are separated (Table 7). There is no interaction between imide groups and N-substituent; hence, the N-substituent did not affect the reduction potential as well as the LUMO level of the investigated compounds. Interestingly, in the cases of 1,5-PhDI and 2,6-PhDI, LUMO orbital localizes on both imide groups; whereas for 1,4-PhDI and 3,3′-PhDI, LUMO orbital localized on one of them. This can be associated with slightly higher angles between imide group and bridge in 1,4-PhDI. In the case of 3,3′-PhDI, the additional twist between naphthalene units in the linker probably also contributes to it (Table 8 and Appendix A).

## 3. Materials and Methods

### 3.1. Materials

Commercially available 1,5-diaminonaphthalene (97%), 1-aminonaphthalene (99+%), 2,6-diaminonaphthalene (95+%) AmBeed, 1,4-diaminonaphthalene (95+%) AmBeed, Benzene-1,2,4,5-tetracarboxylic dianhydride (99%), phthalic anhydride (98+%) and 3,3′-dimethylnaphtidine (95+%) were used as received. Solvents, namely, *N*,*N*-dimethylformamide (DMF; 99.8%), *N*,*N*-dimethylacetamide (DMA; 99.8%), methanol (99.8%), dimethylsulphoxide (DMSO; 99.9%), N-methylpyrrolidone (NMP; 99.5%) and tetrahydrofuran (THF; 99.9+%) were used as received.

### 3.2. Synthesis

#### 3.2.1. Synthesis of 1,5-PhDI

Phthalic anhydride (3 mmol, 355.36 mg) and 1,5-diaminonaphthalene (1 mmol, 157.69 mg) and 1,5 mL of DMA were added to a 10 mL round-bottom flask (Figure 1). The temperature was gradually elevated by 50 °C every 30 min until achieving the boiling point of the solvent (170 °C). The mixture was then refluxed for 24 h. After the reaction had been completed, the mixture was cooled to room temperature and precipitated in methanol. The precipitation was filtered, washed with cold methanol and dried in air, providing a brown-violet crystal solid (337.40 mg, yield 81%).

FTIR (KBr, cm^−1^) υ: 3089–3048 (Ar-H stretching), 1777 (imide assym. C=O), 1713 (imide symm. C=O), 727 (imide ring deformation).

^1^H NMR (600 MHz, CDCl_3_) δ ppm 7.66–7.72 (m, 1 H) 7.75–7.80 (m, 1 H) 7.94–8.04 (m, 2 H) 8.05–8.11 (m, 1 H).

Anal Cald. C_26_H_14_N_2_O_4_ C: 74.64%, H: 3.37%, N: 6.70%, found: C: 75.01%; H: 3.66%; N: 7.18%.

#### 3.2.2. Synthesis of 1,4-PhDI

Phthalic anhydride (3 mmol, 350.98 mg) was dissolved in 1 mL of DMA and heated to 50 °C in a 10 mL round-bottom flask. Afterwards, 1,4-diaminonaphthalene (1 mmol, 159.59 mg) was added and the mixture temperature was rapidly increased to the boiling point of the solvent (170 °C) and refluxed for 6 h (Figure 2). After cooling to room temperature, the mixture was precipitated in methanol. The precipitation was filtered, washed with cold methanol and dried in air, providing a brown crystal solid.

FTIR (KBr, cm^−1^) υ: 3097–3067 (Ar-H stretching), 1779 (imide assym. C=O), 1726 (imide symm. C=O), 716 (imide ring deformation).

^1^H NMR (600 MHz, CDCl_3_) δ ppm:7.60–7.65 (m, 1 H) 7.85 (m, 1 H) 7.93–7.97 (m, 1 H) 8.00 (dd, *J* = 5.46, 3.20 Hz, 2 H) 8.08 (dd, *J* = 5.27, 3.01 Hz, 2 H).

Anal Cald. C_26_H_14_N_2_O_4_ C: 74.64%, H: 3.37%, N: 6.70%, found: C: 74.52%; H: 3.30%; N: 6.51%.

#### 3.2.3. Synthesis of 2,6-PhDI

Phthalic anhydride (3 mmol, 346.98 mg) was dissolved in 1 mL of DMA and heated to 50 °C in a 10 mL round-bottom flask. Afterwards, 1,4-diaminonaphthalene (1 mmol, 159.59 mg) was added and the mixture temperature was gradually elevated by 50 °C every 30 min until achieving the boiling point of the solvent (170 °C) and refluxed for 24 h (Figure 3). After cooling to room temperature, the mixture was precipitated in methanol. The precipitation was filtered, washed with cold methanol and dried in air, providing a brown crystal solid.

FTIR(KBr, cm^−1^) υ: 3090–3045 (Ar-H stretching), 1777 (imide assym. C=O), 1716 (imide symm. C=O), 722 (imide ring deformation).

^1^H NMR (600 MHz, CDCl_3_) δ ppm: 7.66–7.72 (m, 1 H) 7.74–7.79 (m, 1 H) 7.94–8.01 (m, 2 H) 8.01–8.04 (m, 1 H) 8.04–8.10 (m, 1 H).

Anal Cald. C_26_H_14_N_2_O_4_ C: 74.64%, H: 3.37%, N: 6.70%, found: C: 74.75%; H: 3.47%; N: 6.71%.

#### 3.2.4. Synthesis of 1-PMDI

Benzene-1,2,4,5-tetracarboxylic dianhydride (1 mmol, 219.26 mg) was dissolved in 2.75 mL of DMA and heated to 50 °C in a 10 mL round-bottom flask. Afterwards, 1-aminonaphtalene (2.5 mmol, 348.85 mg) was added and the mixture temperature was gradually elevated by 50 °C every 30 min until achieving the boiling point of the solvent (170 °C) and refluxed for 6 h (Figure 4). After cooling to room temperature, the mixture was precipitated in methanol. The precipitation was filtered, washed with cold methanol and dried in air, providing a yellow crystal solid (438.38 mg, yield 94%).

FTIR(KBr, cm^−1^) υ: 3105–3018 (Ar-H stretching), 1776 (imide assym. C=O), 1723 (imide symm. C=O), 724 (imide ring deformation).

^1^H NMR (600 MHz, CDCl_3_) δ ppm:7.57–7.62 (m, 2 H) 7.63–7.68 (m, 2 H) 7.71–7.75 (m, 2 H) 7.76–7.79 (m, 2 H) 7.89 (d, *J* = 8.28 Hz, 1 H) 7.94 (d, *J* = 8.28 Hz, 1 H) 8.12 (d, *J* = 8.28 Hz, 2 H) 8.18 (d, *J* = 7.91 Hz, 2 H) 8.53 (d, *J* = 3.76 Hz, 2 H).

Anal Cald. C_30_H_16_N_2_O_4_ C: 76.92%, H: 3.44%, N: 5.98%, found: C: 76.33%; H: 3.37%; N: 5.82%.

#### 3.2.5. Synthesis of 3,3′-PhDI

Phthalic anhydride (3 mmol, 342.64 mg) was dissolved in 3 mL of DMA and heated to 50 °C in a 10 mL round-bottom flask. Afterwards, 3,3′-dimethylnaphtidine (1 mmol, 314.64 mg) was added and the mixture temperature was gradually elevated by 50 °C every 30 min until achieving the boiling point of the solvent (170 °C) and refluxed for 6 h (Figure 5). After cooling to room temperature, the mixture was precipitated in methanol. The precipitation was filtered, washed with cold methanol and dried in air, providing a brown crystal solid (418.32 mg, yield 73%).

FTIR(KBr, cm^−1^) υ: 3068–3027 (Ar-H stretching), 2925 (C-H aliphatic stretching), 1781 (imide assym. C=O), 1723 (imide symm. C=O), 724 (imide ring deformation).

^1^H NMR (600 MHz, CDCl_3_) δ ppm: 2.37 (s, 3 H) 7.35 (d, *J* = 8.28 Hz, 1 H) 7.45 (t, *J* = 7.72 Hz, 1 H) 7.55 (t, *J* = 7.72 Hz, 1 H) 7.73 (s, 1 H) 7.85 (d, *J* = 8.66 Hz, 1 H) 8.00–8.06 (m, 2 H) 8.09–8.16 (m, 2 H).

Anal Cald. C_38_H_24_N_2_O_4_ C: 79.71%, H: 4.22%, N: 4.89%, found: C: 79.15%; H: 4.18%; N: 4.78%.

### 3.3. Characterisation Methods

^1^H-NMR spectra of synthesized polyazomethines have been recorded on an Avance IIUltrashield Plus spectrometer, operating at 600 MHz, using deuterated chloroform (99.95%) as a solvent and tetramethylsilane (TMS) as an internal reference. The FTIR spectra have been recorded on a JASCO FTIR 6700 Fourier transform infrared spectrometer, in a transmittance mode, in the range of 4000–400 cm^−1^ at a resolution of 2 cm^−1^ and for 64 accumulated scans. DSC measurements have been taken with a DSC Q2000 apparatus (TA Instruments, Newcastle, DE, USA), in a range of −50–380 °C under the nitrogen atmosphere (flow rate was 50 mL/min), using aluminum sample pans. The instrument has been calibrated with a high-purity indium. In this study, the glass transition temperature (T_g_) has been taken as a midpoint of heat capacity change for amorphous samples obtained by quenching from the melt in liquid nitrogen. Thermogravimetric analysis (TGA) has been performed with TGA/DSC1 Mettler Toledo thermal analyses, in a range of 25 to 600 °C at a heating rate of 10°/min in a stream of nitrogen (60 mL/min). The obtained TGA data have been analyzed with the Mettler Toledo Star System SW 9.30. The absorption spectra of solutions have been recorded in ranges of 270–800 nm (DMF). Concentration of all investigated solutions was 1 × 10^−4^ M.

Electrochemical analysis (cyclic voltammetry, CV; differential pulse voltammetry; DPV) was performed in a three-electrode system using an electrochemical cell equipped with the platinum working electrode, the platinum auxiliary electrode in the form of a coil and the silver pseudo-reference electrode, using an Eco Chemie (Utrecht, the Netherlands) Autolab M101 potentiostat. The potential of the pseudo-reference electrode was calibrated versus the ferrocene/ferrocenium (Fc/Fc+) redox couple. An amount of 0.1 M solution of tetrabutylammonium hexafluorophosphate (Bu_4_NPF_6_; 98+%) in dichloromethane (DCM; HPLC grade) or *N*,*N*-dimethylformamide (DMF; HPLC grade) serves as a supporting electrolyte. Electrochemical analysis was conducted for 1 × 10^−3^ M solutions of investigated compounds dissolved in the previously prepared electrolyte. The CVs and DPVs were registered using a scan rate of 0.1 V/s. In the case of CV, the step potential was equal to 0.00244, whereas DPVs were registered using the step potential of 0.01 V and the modulation amplitude of 0.07 V.

Spectroelectrochemical measurements were carried out using a UV-Vis Hewlett Packard (Palo Alto, CA, USA) 8453 spectrophotometer and JEOL (Tokyo, Japan) JES-FA 200, an X-band CW-EPR spectrometer operating at 100 kHz field modulation and an Cypress Systems (California, USA) OMNI 90 or Eco Chemie (Utrecht, The Netherlands) AUTOLAB PGSTAT100N potentiostat.

The UV–Vis spectroelectrochemical cell made of a quartz cuvette was equipped with the ITO working electrode, an auxiliary electrode made of platinum, and a silver pseudo-reference electrode. The EPR spectroelectrochemical cylindrical cell consists of the platinum wire as a working electrode, the platinum spiral as an auxiliary electrode and the silver wire as a pseudo reference electrode. Microwave power and modulation width were adjusted in each case in order to provide non-saturated and well-resolved spectra. The spectra of the radical anions were registered during the potentiostatic reduction at potentials determined from CV measurements. Simulation of EPR spectra lineshape was conducted with the WinSim software [41].

Spectroelectrochemical analysis was performed for 0.5 and 1.0 mM solutions of investigated compounds dissolved in supporting electrolyte such as in the case of electrochemical measurements. All of the electrochemical and spectroelectrochemical investigations were conducted on argon-purged solutions.

### 3.4. Theoretical Calculation

DFT calculations were carried out with a B3LYP hybrid functional [42,43,44,45] combined with a def2SVP basis set [46]. Ground state and diradical dianion geometry optimalization was followed by frequency calculations; in all cases, no imaginary frequencies were found. All calculations in this work were conducted with a polarizable continuum model (PCM) using dichloromethane as a solvent and were performed with Gaussian 09.E software [47] using PLGrid structure.

## 4. Conclusions

Summarizing, four phthalimides derivatives consists of two imide groups connected via naphthalene (1,5-PhDI, 1,4-PhDI, 2,6-PhDI) or 3,3′-dimethylnaphtidin (3,3′-PhDI) bridge were synthesized and analyzed in depth. Additionally, a compound with the pyromellitic diimide unit capped with two 1-naphtalene substituents was obtained. All synthesized materials undergo electrochemical reduction. The reduction potential is almost the same for all compounds with naphthalene bridge. However, the introduction of 3,3′-dimethylnaphtidine leads to the noticeable increase in the reduction potential compared with the naphthalene unit, confirming the promising properties of polyimides with such a building unit. The pyromellitic diimide derivative is the easiest to reduce out of all the investigated molecules due to the fact that the diimide unit is a stronger electron-acceptor compared with monoimide moiety. The reduction of all compounds leads to the generation of stable radical anions which was confirmed by EPR spectroelectrochemistry. The analysis of the hyperfine structure of obtained EPR spectra allows us to conclude that in all cases, radical anions localize on the imide or diimide core, which is in agreement with theoretical calculations. This also shows that there is no interaction between the imide/diimide unit and N-substituents. To evaluate the potential electrochromic properties of investigated compounds, a UV–Vis spectroelectrochemical experiment was performed. In all cases, electrochemical reduction leads to reversible changes in UV–Vis spectra; however, for imide derivatives, the absorption bands related to the reduced, as well as for neutral form, are located in the UV region. Nevertheless, the introduction of 3,3′-dimethylnaphtidine leads to the noticeable bathochromic shift of the absorption band associated with radical anion generation, leaving the absorption of the neutral form in the UV region. This indicates that even though the bridge between the imide groups has little impact on the reduction potential and optical properties of neutral forms, it is quite important when it comes to the absorption features of reduced forms. This is important due to the fact that the change of the bridge between imide groups can result in the obtaining of cathodic electrochromic materials possessing colorless and colored states, which indicates that the optimization the N-substituted phthalimide structure allows us to obtain stable, cathodic electrochromic materials. Similarly, in the case of the pyromellitic diimide unit, which has been rather overlooked as a building block for electrochromic materials, the obtained results clearly showed that the proper choice of N-substituents can improve the electrochromic response of pyromellitic diimide derivatives.

## Data Availability

Data are presented in the article and Appendix A.

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
