# Peer review of "Naphthalene Phthalimide Derivatives as Model Compounds for Electrochromic Materials"

_molecules, 2023, doi:10.3390/molecules28041740_

Round 1
Reviewer 1 Report
The authors report the synthesis of 5 phthalamides and their characterization by various techniques, for example, FTIR, NMR, DSC, TGA, CV, UV-Vis. The manuscript, in general, presents many grammatical errors and therefore, it is difficult to comprehend. In addition, it is necessary to significantly improve the manuscript, for this reason he considered major revisions.
Some points to improve
Point 1. The abstract must be reorganized into the following sections:
Introduction; Motivation; Gap; Proposal; Methods; Results; Conclusions
Point 2. Introduction, it is necessary that the authors capture the originality of the work.
Point 3. What is the objective of this study?
Point 4. Experimental section, materials, lack of purity of reagents
Point 5. Place a synthesis subtitle for each reaction.
Point 6. Place FTIR and NMR spectra in supplementary material.
Point 7. Deuterated chloroform or DMSO? Line 156.
Point 8. Line 174. Silver pseudo reference electrode. Or Ag/AgCl as reference electrode?
Point 9. The results and discussion / materials and methods is very confusing in terms of the computational details:
b) They dot not mention the computational packaged employed for this simulation, e.g., Gaussian, Gamess, ORCA, NWChem, and the citation is missing. It is also important to mention which version of the code was employed.
Point 10. The DSC and TGA are missing from the manuscript, without this I cannot ask questions until the second review.
Point 11. Please further discuss results in optical properties, that other authors have reported these spectra?
Point 12. The CV voltammogram of 1-PMDI (Figure 2a) corresponds to a known mechanism of arylene diimides reduction. Are they not mentioned by those who have been studied?
Point 13. Because the working electrode and counterelectrode are platinum. Do they have the same area and shape?
Point 14. These results are very interesting, there are works studying the properties Spectroelectrochemical UV-Vis?
Point 15. Broaden the discussion of theoretical results
Point 16. Improve conclusions
Reviewer 2 Report
In my opinion the article is interesting.
The introduction is providing enough citations. The research scope is mentioned - cathodic coloring electrochromic materials, and the limitations of current existing materials. There is a small error to correct: line 41 "H2O/O2 - complex" replace with: "H2O/O2 - complex"
Experimental:
- at 2.1. Materials - there are some substances for which the manufacturer is not mentioned
- Line 171 - for Electrochemical analysis (cyclic voltammetry; CV and differential pulse voltammetry; DPV) there are some parameters not mentioned: scan rate, step, potential domain, amplitude
Results:
- it could be useful to add the NMR and FTIR spectra to supplementary material
- Figure 1 - in the figure 1 PhDI appears with red line, but in text 1-PMDI is mentioned - need to be corrected
- line 269 "effectiveπ-conjugation area" need to be replaced by : "effective π-conjugation area"
Conclusions: Line 401 - 404 - " The analysis of hyperfine structure of obtained EPR spectra allows to conclude that in all cases radical localize on imide or diimide core which is in agreement with theoretical calculation, which shows that there is no interaction between imide/diimide unit and N-substituent." The phrase is to long and must be reformulated.
Round 2
Reviewer 1 Report
The authors heeded the suggested recommendations, I recommend their publication.